# What a Scientific Language Model Knows and Doesn't Know about Chemistry

**Lawrence Zhao**
Yale University
larry.zhao@yale.edu

**Carl Edwards**
University of Illinois
cne2@illinois.edu

**Heng Ji**
University of Illinois
hengji@illinois.edu

## Abstract

Large Language Models (LLMs) show promise to change how we can interact with and control the design of other modalities, such as drugs, materials, and proteins, and enable scientific reasoning and planning. However, LLMs have several weaknesses: they tend to memorize instead of understand, and the implicit knowledge does not always propagate well between semantically similar inputs. In this work, we seek to distinguish what these scientific LLMs have memorized versus what they actually understand. To do so, we propose a new comprehensive benchmark dataset to evaluate LLM performance on molecular property prediction. We consider Galactica 1.3B, a state-of-the-art scientific LLM, and find that different prompting strategies exhibit vastly different error rates. We find that in-context learning generally improves performance over zero-shot prompting, and the effect is twice as great for computed properties than for experimental. Furthermore, we show the model is brittle and relies on memorized information, which may limit the application of LLMs for controlling molecular discovery. Based on these findings, we suggest the development of novel methods to enhance information propagation within LLMs—if we desire LLMs to help us control molecular design and the scientific process, then they must learn a sufficient understanding of how molecules work in the real world.

## 1 Introduction

Chemistry is a domain with a scarce amount of labeled data [1]. Many chemical properties don't have entries in human constructed knowledge bases, and many entities don't have enough properties. This is because it is often difficult to collect molecular data without direct experimentation, which is time and cost-expensive. Therefore, in this paper, we explore molecular property prediction using computational tools, potentially bypassing the need for additional experimentation. Certain properties, such as molecular weight and rotatable bond count, can be easily computed by a program given a chemical's structure. Other properties may only be truly verified by experimentation. In this paper, we refer to these properties as "computed properties" and "experimental properties," respectively.

Large language models (LLMs) have emerged as an effective tool for making predictions about new molecules [2, 3]. While certain tools like search engines can only retrieve information from existing sources of knowledge, machine learning approaches are capable of predicting properties for unknown molecules. Large language models have the potential to leverage their knowledge and reasoning abilities for molecular representation learning and thus make deeper insights. LLM knowledge can be described in two categories: parametric knowledge of specific facts that the LLM repeats from training data (which we refer to as "memorized") and implicit knowledge that allows the model to evaluate unseen entities (which we refer to as "understanding") [4].

Beyond simple property prediction, however, natural language can also provide a useful way to acquire knowledge about molecules [5]. This is in large part because language has been developed

over the centuries for scientists to propagate knowledge about, reason, and understand molecules. Through a language model, we can control molecules and their properties at a high level and bridge the gap between drug names, molecules, and their properties. Further, LLMs hold promise for performing scientific reasoning and delegating experiments to automated laboratories [6]. If we want them to live up to this potential, they need to have some degree of understanding, rather than memorization. An ability for language models to understand molecular structure is important for making accurate statements about molecules.

In our study, we represent molecules textually and give them to the model as as SMILES strings [7,8]. SMILES is the most prevalent method of text-based linear representation of chemical structure in cheminformatics. This generally means that LLMs have seen SMILES in training. While other more rigorous and self-consistent representation methods like SELFIES [9] have been created, current models do not seem to perform significantly better when trained on SELFIES as opposed to SMILES [10] on various downstream tasks. Currently, the best models that learn molecular representations are not solely text-based and include some additional geometrical information. Some models are multimodal (e.g., Uni-Mol [11]), allowing additional three-dimensional molecular information to be encoded, while others unify molecule structure encoded as a graph and textual information in a single language model (e.g., GIMLET [12]). Some recent work leverages chemical reactions as additional context to condition molecule representations [13]. While one can argue that additional information is required to make the most accurate property predictions, SMILES strings are interchangeable with graphical representations and possess sufficient implicit information for a language model to make initial predictions. Although the same molecule might map to multiple valid SMILES string representations, a SMILES string corresponds to a unique molecule.

To summarize, our main contributions include:

1. We evaluate the discrepancy between memorization and understanding of molecules as represented by SMILES in Galactica.
2. We release a new benchmark dataset for evaluating this discrepancy across many properties.
3. We investigate regression accuracy in Galactica across multiple prompting styles. This allows us to identify structural similarity and textual proximity as key factors that influence which values are copied from context, affecting performance of in-context learning.

## 2 Related Work

Modern Large Language Models (LLMs) generally consist of a large transformer-based architecture [14] pretrained via some form of masked language prediction on a large corpus of unlabeled data. These models are often fine-tuned to perform specific downstream tasks. One notable aspect of LLMs is that they have been shown to memorize their training data [15]. In-context learning (ICL) has been used to enhance the performance on scientific tasks without the need for additional training or modification. Previous work has found that randomly replacing labels in ICL barely hurts performance on a range of classification and multi-choice tasks [16].

**Scientific LLMs**  Following the strong performance of language models such as BERT [17], it was quickly uncovered that domain-specific scientific pretraining could offer exciting gains in performance on scientific downstream tasks [18, 19]. Currently, state-of-the-art scientific domain large language models, such as Galactica [20] and BioGPT [21], are autoregressive and extend ideas behind models such as GPT-3 [22]. In this work, we choose to focus on Galactica because it has been explicitly trained on molecular representations in the form of SMILES strings [7, 8]. These strings fully represent molecular graphs, ergo they contain sufficient information for molecular property prediction tasks. Galactica explicitly tokenizes these SMILES strings by wrapping them inside specialized tokens. To train the model, Galactica pretrains on a large scientific corpus consisting of data from knowledge bases like PubChem and scientific research papers. Galactica was shown capable of predicting molecule IUPAC names from SMILES strings in a self-supervised manner, although notably behavior peaks at a high parameter size (120 billion) and is limited at lower parameter sizes.

**Evaluating LLMs on Chemistry**  Due to the capabilities of LLMs, several studies have attempted to evaluate the capabilities of general domain models released by OpenAI, such as GPT-3 [22], regarding knowledge of chemistry [23, 24]. Past work has evaluated several LLMs for molecular property prediction in ICL, but they measured accuracy of LLMs on a small subset of popular

benchmark binary classification datasets [10, 25]. [25] determined that ICL is always beneficial. However, binary classification is relatively easy for LLMs and this may not be an accurate reflection of what the model "knows." While the results of LLMs on chemistry are impressive, in our work we instead evaluate the difference between actual understanding versus memorization and copying.

**Language+Molecules** While Galactica is a notable example of a large-scale science-focused multimodal LLM, work had already been done to adapt LLMs to language and molecules [26–29]. Notable tasks include cross-modal molecule retrieval [26, 28], cross-modal translation [27, 30, 31], molecule editing [32], assay activity prediction [33], and connecting several biochemical modalities [34]. This body of work has helped uncover exciting possibilities for combining language with chemistry—however, for further progress it is desirable to pretrain strong foundational models which understand instead of memorize. In this work, we attempt to measure this shortcoming, which will help inform future efforts in this field.

**Machine Learning for Property Prediction** Descriptive and graphical neural networks have been applied for molecular property prediction [35]. Further, benchmark tasks for property prediction have emerged to compare these models [36]. Language models trained solely on molecules and finetuned on these benchmarks have also arisen as strong predictors of molecule properties [3]. Given the growing interest in using LLMs to control chemical reasoning and design [5], the ability to use natural language-based LLMs for property prediction is a clear and critical next target. Since LLMs are expensive to train, it is also desirable to update their abilities after pretraining. Further, it is well-known that injected information does not propagate well in LLMs [37]. Thus, it is desirable to evaluate LLM's memorization and ability to propagate information. However, to the best of the authors' knowledge, there is no benchmark dataset to accomplish this. To address this gap, we propose a benchmark for evaluating on diverse group of properties in a variety of prompt styles.

## 3 Methods

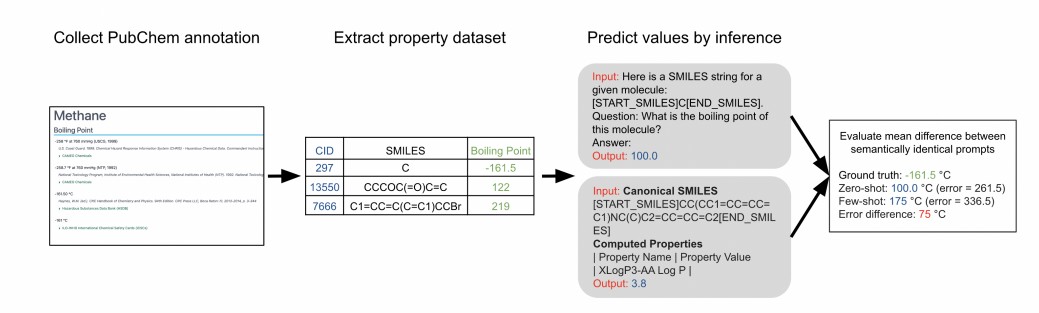

Figure 1: Overview of the project workflow. We construct a dataset by scraping PubChem records for each property. Prompting Galactica generates predictions that can be compared with ground truth values in the dataset.

### 3.1 Dataset construction

We selected 35 properties to create a comprehensive dataset of chemical property prediction tasks. These properties include experimental properties, such as density and heat of vaporization, and computed properties, such as molecular weight and Log *P* (which is a measure of a compound's lipophilicity). First, we scrape all SMILES-value pairs for a given property from public PubChem Compound records using the PubChem PUG View API. Out of these examples, we randomly set aside 100 pairs for evaluation and set the other examples aside for ICL. For each of these properties, we have proposed a variety of classification and regression tasks depending on the nature of data available for these properties. If there are multiple potential values corresponding to a single CID in PubChem, only one value is randomly selected. See Table 4 for dataset composition and Fig 3 for prompts.

## 3.2 Evaluation

The Galactica model with 1.3 billion parameters was used for all inference tasks. For each compound (as represented by SMILES string), we chose 5 random examples (k=5) for ICL. We also test whether the values of the context examples matter; for the random experiments, we assign both randomly generated values (-3 to 7 for Log $P$, 0 to 1000 for molecular weight) to each string. The model was evaluated for 100 compounds in each experiment. The evaluation set for each property was a random sample of all available data points unless otherwise specified. After inference, extra tokens following the generated value were removed. Top-1 accuracy under greedy search decoding was used, as in [20]. We note that, because massively-pretrained scientific LLMs have been exposed to a wide corpus, there is almost certainly content overlap between Galactica's pre-training corpus and our dataset. Since we are interested in investigating memorization, this is not an issue.

## 4 Results

For each of the chemical properties we have selected, we are interested in the model's performance in zero-shot and few-shot scenarios to distinguish what the model has already "memorized" and what it can learn. If the model can learn in-context, we also consider this "understanding" in addition to zero-shot. The below experiments on regression tasks and statistical analysis support the idea that memorized parametric knowledge plays a large role in model output. For each experiment, we plot predicted values against ground truth values.

### 4.1 Prompting strategy creates significant variance in model performance.

As a first step, we evaluate the model's ability to predict two properties: the partition coefficient (Log $P$), which is a measure of lipophilicity, and molecular weight. Log $P$ and molecular weight are both important for drug discovery. Galactica has been pre-trained on these properties in a table format, which makes them amenable to testing for memorization. We evaluate these properties on two sets of 100 compounds—one set of compounds are the first 100 molecules on PubChem, which the model has likely seen before, while the next set of compounds are randomly selected from a range of 100 to 100 million. For both of these sets, we investigate the effect of prompting strategy on regression quality. We notice that the errors produced by the randomly selected compounds tend to be lower across the board than the first 100 compounds. Thus, the difference in error between the two sets of molecules cannot be attributed to the earlier compounds being more "memorized" than the others.

Based on Table 1, overall trends in prompting differences are consistent within each property, regardless of the evaluation compound set. For molecular weight, ICL with 5 examples seems to achieve lower accuracy than even zero-shot prediction in some cases. Based on previous studies of LLM behavior [16], using random values for context examples would be expected to result in roughly the same error as true values in examples. However, ICL with random values substantially lowers $r^2$ (indicating worse regression quality) and increases RMSE. Both of these indicators suggest that Galactica does better when it can use prior memorized knowledge to make predictions rather than learning input-text mappings through ICL for molecular weight. While Galactica has some predictive power for these properties, it seems on the other hand that the model is unable to predict Log $P$ with

Table 1: Effect of prompting strategy on predictive accuracy of computed properties (RMSE), Galactica 1.3 billion parameter model. The table format prompt used was used in pre-training Galactica. We also evaluated with "few-shot (true)" prompts, which included examples of SMILES and their ground truth values, and "few-shot (random)" prompts, which used randomly generated values instead of ground truth.

| Property | Table | Zero-shot | Few-shot (true) | Few-shot (random) |
|---|---|---|---|---|
| Molecular Weight (first 100) | 150.69 | **120.51** | 225.07 | 454.66 |
| Log $P$ (first 100) | **2.501** | 4.252 | 5.362 | 4.654 |
| Molecular Weight (random) | 122.6 | **96.28** | 126.49 | 243.86 |
| Log $P$ (random) | **1.537** | 3.372 | 2.096 | 3.912 |

good accuracy outside of the memorized table context. Thus, the ability for Galactica to predict Log $P$ is more brittle to prompt selection than molecular weight. This difference in trends might be because although both molecular weight and Log $P$ were included in the pre-training computed property prompts, we hypothesize molecular weight was seen more often in the training data, and the sequence for computed Log $P$, "XLogP3-AA", might have only occurred in the table context.

We find that even prompts that are almost exactly the same produce wildly different error rates. For example, we observed that merely changing the label of "XLogP3-AA Log $P$" to simply "Log $P$" when prompted in the table format increased root mean squared error exhibited by the model on molecules 1-100 by 50%. The success of generating memorized information is highly dependent on the context that we provide the model, even if the core question is semantically identical.

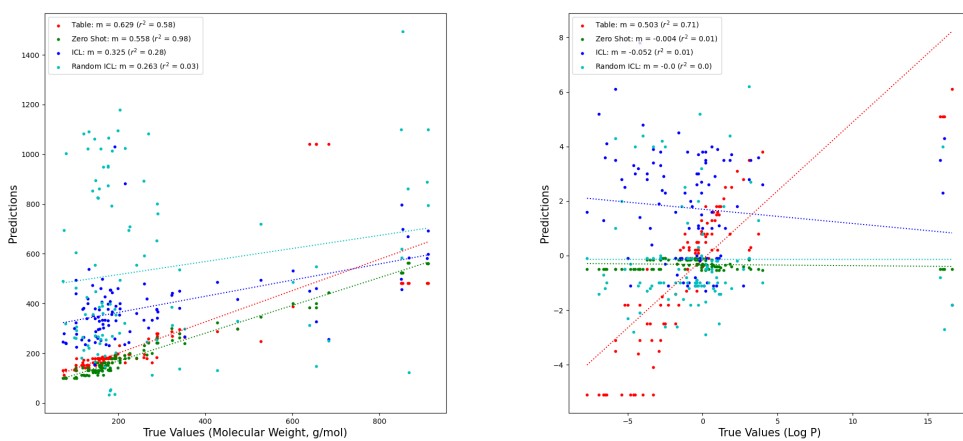

(a) Predicted vs true values of molecular weight on PubChem CID 1-100.

(b) Predicted vs true values of Log $P$ on PubChem CID 1-100.

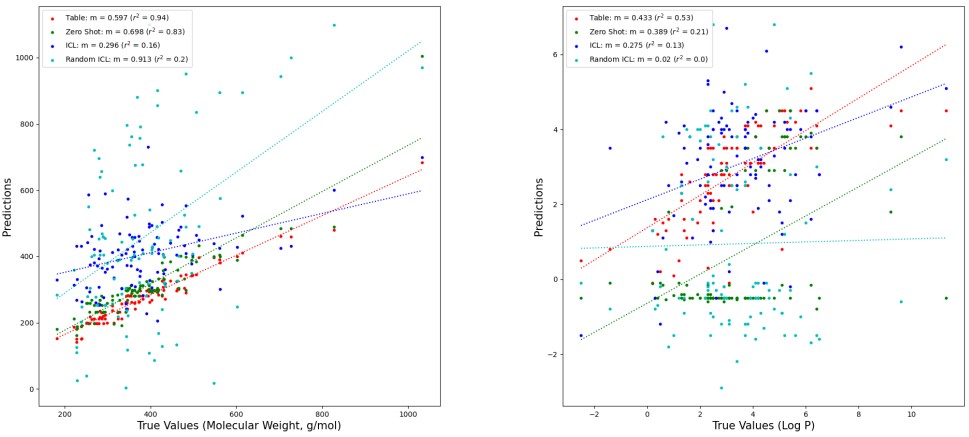

(c) Predicted vs true values of molecular weight on random CIDs.

(d) Predicted vs true values of Log $P$ on random CIDs.

Figure 2: Regression plots of predicted vs true values for molecular weight and Log $P$. $m$ is the slope of best fit. Plots (a) and (c), which are for molecular weight, share a similar drop in $r^2$ from zero-shot and table prompts to few-shot prompts. Plots (b) and (d), which are for Log $P$, show that the table prompt has a relatively strong $r^2$ value compared to other prompting strategies. This indicates that for Log $P$, memorized knowledge is only recalled when prompted in the table format.

## 4.2 Galactica copies values provided in-context, leading to discrepancies between few-shot and zero-shot.

The absolute percent RMSE difference between zero-shot and few-shot learning prompts, on average, is 27%. At a surface level, ICL "spreads out" the distribution of the predicted values from the corresponding zero-shot predicted values across all properties. In all cases, we observe that the model generally underpredicts the ground truth value.

ICL functions differently than zero-shot for property prediction. When Galactica has memorized information about a property, Galactica struggles to reconcile contextual information with memorized parametric knowledge when it is present. When the memorized parametric knowledge cannot be accurately recalled in a zero-shot context, few-shot prompts containing ground truth examples will score higher on regression tasks than zero-shot prompts, since the model is able to use the new context to form future predictions. Interestingly, we observe that, as shown in Table 2, ICL tends to "copy" the values from the context examples when outputting the value for the prompted compound. We can only quantify "copying" for properties that lie on a continuous distribution. Properties like hydrogen bond count are discrete variables, so it is difficult to distinguish a valid prediction vs a copied value. On the other hand, it is very unlikely for two compounds to have the same value on a large continuous distribution, so when compounds share values, it is extremely probable the model has copied an in-context SMILES label and attributed it to the prompted SMILES string.

**Copying is not necessarily counterproductive** Surprisingly, copying values from the ICL examples does not appear to have a significant negative effect on predictive quality. Copying values might be expected to reduce accuracy. However, we find that average error among copied examples is greater than average error among non-copied example for only 4 out of 8 properties. We observe the model disproportionately copies the values of SMILES strings that are more structurally and textually similar than the average example in the context. The event that a SMILES string occurs at the end of the context and the event that a SMILES string is syntactically similar to the query molecule are fundamentally independent.

**The model repeats values from context SMILES that are structurally similar to the queried molecule.** Levenstein distance between the SMILES string of the copied-from example and the prompted SMILES serves as a good metric to estimate the extent to which similar molecules are copied. We compare index of the copied SMILES among examples SMILES ranked by this edit similarity with SMILES in prompt vs expected rank. We would expect this to be position 3 (the positions range from $1 - 5$) if the model was copying strings purely independently of position in the example list (this is the null hypothesis). However, the average index is $2.68$ across all properties. A right-tailed chi-squared test reveals $\chi^2 = 0.64$, p value $= 0.001$, so we reject the null hypothesis. This result is consistent with LLMs having a weak ability to associate molecules in its context that possess similar structures. New methods to improve knowledge propagation would enhance this ability.

Table 2: Experimental and computed properties inference on the 1.3 billion parameter Galactica model, $n = 100$. % copied refers to the percentage of copied examples in few-shot inference.

| Property | Zero-shot (RMSE) | Few-shot (RMSE) | (% copied) |
|---|---|---|---|
| Log $P$ | 3.372 | **2.096** | 57 |
| Molecular Weight | **96.28** | 126.49 | 82 |
| Hydrogen Bond Donor Count | **1.371** | 1.559 | N/A |
| Hydrogen Bond Acceptor Count | 5.239 | **4.650** | N/A |
| Atom Stereocenter Count | **1.507** | 2.742 | N/A |
| Topological Polar Surface Area | 98.82 | **52.64** | 52 |
| Density | 3.368 | **2.691** | 70 |
| pK$_a$ | 10.62 | **8.316** | 33 |
| Melting Point | **361.10** | 366.65 | 52 |
| Boiling Point | 858.65 | **766.75** | 55 |
| Heat of Vaporization | 150.78 | **74.59** | 46 |

**The model repeats values previously seen towards the end of context.** Upon a closer examination of the results of the ICL experiment, Galactica tends to parrot the context values at an alarmingly high rate (about $60\%$, see Table 3). In particular, it tends to repeat the last value provided in the context portion of the prompt as the output for ICL—41% of the time! This is consistent with increased attention for the tokens at the beginning and end of the context window, as explained in [38]. We speculate pre-training causes a recency bias since tokens are correlated with their nearby tokens (Firth's "you shall know a word by the company it keeps").

### 4.3 Differences between computed properties and experimental properties

We evaluate whether performance in the ICL setting vs. zero-shot setting is different between computed and experimental properties. Overall, we find that ICL is generally better for both computed and experimental properties, but that ICL is roughly twice as good on computed than for experimental. Experimental and computed results are statistically distinct with $p < 0.05$ (see Appendix C). However, we note that molecular weight is an exception to this trend, as the $r^2$ and RMSE values for molecular weight indicate that regression quality significantly worsens from zero-shot to few-shot.

There are several explanations for the increased effect of ICL on computed properties compared to experimental properties. Galactica has already been explicitly pre-trained on computed properties, while data is much more sparse for experimental properties so the model has seen less information about these properties. Furthermore, experimental properties require a deeper understanding of the interaction between a molecule and the outside world, making them fundamentally more difficult to predict than computed properties. Essentially, the model does not have enough parametric information to understand the association between SMILES and values for experimental properties. Additionally, if the model has already memorized information about a SMILES string, other SMILES strings in the context may "confuse" the model, which explains the performance degradation for molecular weight.

### 4.4 Evidence of memorization

**Repeating the training data** In the case of the zero-shot predictions for molecular weight, the model generates entire computed properties tables after it has generated the correct value. In other cases, the model outputs additional properties or a description of what the property is, which was seen in the training data (see Appendix E for examples). Also, when no additional context is generated, zero-shot predictions tend to produce a default value that is less accurate. These observations suggest the model has associated SMILES strings with the context around them in the training data rather than by understanding the correct output. Memorization around context is the dominant mode of learning.

**The model generates unrealistic values** While the model produces numerical values for all properties when prompted, some of the values suffer from hallucination and are not realistic. As a striking example, we observe negative outputs for density. As another, Galactica cannot distinguish between discrete and continuous values, since it will output non-integer values for hydrogen bond count in the zero-shot setting.

### 4.5 Discussion

Why might Galactica fail to provide accurate answers to our prompts? We can point to several external reasons relating to the model's design that might lead to significant knowledge gaps. First, while PubChem documents were included in the training data, these were mixed with other data sources such as papers in the training corpus, which may confuse the model due to inconsistency in

Table 3: The proportion of examples copied from index $i$ (denoted as head). Results are combined from all ICL experiments of continuous variables that use true values in examples. The average index among all repeated examples is $4.3$.

| 1 | 2 | 3 | 4 | 5 | Total |
|---|---|---|---|---|---|
| 0.038 | 0.037 | 0.039 | 0.076 | 0.407 | 0.597 |

the scientific literature. Second, PubChem is not necessarily a representative training dataset. There are biases in which types of molecules are included in PubChem (e.g., towards smaller, drug-like organic molecules). Given these biases, a better way of generating new SMILES with a range of chemical features is needed to serve as an evaluation dataset.

We found that LLMs suffer from popularity bias, in which the LLM may generate a token that appears often in the training corpus. Popularity bias is not ideal for property prediction tasks, for which one answer or result may be significantly more prevalent in the data. This is because, for the model to possess an implicit understanding of molecules, it should be able to learn molecular features that allow it to predict values for outliers, rather than memorizing the most common value. Furthermore, chemical property prediction requires precise answers for meaningful applications. While hallucinations may help the model produce coherent output for unfamiliar inputs, slight differences in numerical values can be incredibly significant, especially for values reported on a logarithmic scale (e.g., $pK_a$).

To improve molecular understanding, we also attempted fine-tuning the 125 million parameter Galactica model to predict boiling points. However, this was met with limited success ($r^2 = 0.06$ on validation), which is less than the value of boiling point in the few-shot setting ($r^2 = 0.19$). Furthermore, we observe catastrophic forgetting for molecular weight prediction following the fine-tuning procedure. Hence naive fine-tuning is insufficient to adapt Galactica to downstream chemical property prediction tasks.

## 5  Conclusions

LLMs have shown the potential to tackle challenges that require significant molecular understanding. However, it is unclear the extent to which these abilities are currently limited by LLM memorization. To investigate this problem, we evaluated the effect of prompting strategy on error and regression quality on a wide range of properties for Galactica. The high rate which context example values are copied for model predictions during in-context learning is consistent with the model predominantly encoding knowledge in its parameters, rather than by "understanding." Furthermore, we found general problems in LLMs for molecules that would make these prediction tasks challenging. Currently, all LLMs suffer from popularity bias and hallucination. Also, LLMs struggle to understand numbers, which makes them perform poorly at regression tasks.

We propose several ways to fix these issues. First, it may be worth exploring the conversion of numbers into range tokens during pretraining. This might accurately reflect the nature of chemical data, since results are often reported as a range of values, and resolve the problem of literature inconsistency. Greedy decoding (as used by Galactica) is a poor approach for producing numerical values, since the relative size of a number depends on all digits and their relative positions. It may be interesting to develop new decoding methods to produce accurate quantitative values. Second, given that fine-tuning performs poorly and causes catastrophic forgetting, we hope to develop and implement novel methods to facilitate injected knowledge propagation in LLMs to improve property prediction. These efforts build on previous work to successfully update knowledge in LLMs [39] or fuse new action knowledge into a frozen LLM [40]. Improvements like this may improve knowledge propagation in LLMs, giving them a deeper understanding of molecule representations and improving property prediction on novel compounds. The resulting improved LLM understanding of molecules will be critical to the use of LLMs to accelerate the process of designing and testing new molecules (e.g., in drug discovery).

**Limitations**  It is difficult to directly measure understanding vs. memorization of abstract concepts in any LLM. There have been methods that quantify this information [15], but without direct access to the training corpus, it is difficult to directly quantify how much is memorized. Here, we use the "brittleness" of the LLM as a proxy for measuring LLM memorization. Essentially, we are able to evaluate when the LLM has memorized a property by comparing its performance with one prompt style (e.g., table formatting) versus another (e.g., zero-shot). We note that model parameter size may change the observed results, since larger LLMs tend to have emergent reasoning abilities at higher parameter sizes.

## Acknowledgments

This research is based in part upon work supported by the Molecule Maker Lab Institute: an AI research institute program supported by NSF under award No. 2019897. The views and conclusions contained herein are those of the authors and should not be interpreted as necessarily representing the official policies, either expressed or implied, of the U.S. Government. The U.S. Government is authorized to reproduce and distribute reprints for governmental purposes notwithstanding any copyright annotation therein.

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

# A  Dataset details

PubChem properties were selected that satisfy the following criteria:

- $n \geq 500$ (n = number of of unique SMILES included) so that there was sufficient data in the training data
- It would be plausible to evaluate model performance compared to real-world data in a meaningful way
- Units were standardized across molecules, and that information was included in the prompt if not implicitly needed— (i.e. molecular weight is implicitly understood to be given in g/mol, while all boiling points were expressed and evaluated in degrees Celsius, density in g/mol, Henry's law at room temperature).
- All properties evaluated under standard conditions unless otherwise specified.
- For classification tasks, we picked the most common values and then used molecules under those classes to construct the dataset.
- Specific prompts for each property will be made available in the GitHub.

Table 4: Dataset composition of selected PubChem properties and their respective tasks. $n$ refers to the number of unique CIDs present on PubChem or included in the dataset for a certain property. TBD $n$ (dataset) values refer to a value whose dataset have not yet been extracted yet. Data current as of September 28, 2023.

| | Task | | |
|---|---|---|---|
| Type | Property | $n$ (PubChem) | $n$ (Dataset) |
| Regression | Molecular Weight | ~116 million | 10000 |
| Regression | Log $P$ (predicted) | ~All | 10000 |
| Regression | Topological Polar Surface Area | ~116 million | 10000 |
| Regression | Hydrogen Bond Donor Count | ~116 million | 10000 |
| Regression | Hydrogen Bond Acceptor Count | ~116 million | 10000 |
| Regression | Rotatable Bond Count | ~116 million | 10000 |
| Regression | Stereocenter Count | ~116 million | 10000 |
| Regression | Autoignition Temperature | 946 | 679 |
| Regression | Boiling Point | 6130 | 1807 |
| Classification | Chemical Classes | 18169 | 10598 |
| Regression | Collision Cross-Section | 18169 | TBD |
| Classification | Color | 6175 | TBD |
| Classification | Form | 6175 | TBD |
| Classification | Corrosivity | 698 | TBD |
| Regression | Density (g/mL) | 5831 | 1159 |
| Regression | Dissociation Constants (pK$_a$) | 2571 | 471 |
| Regression | Flash Point (Closed Cup) | 2158 | 1376 |
| Regression | Heat of Combustion | 500 | 126 |
| Regression | Heat of Vaporization | 708 | 320 |
| Regression | Henry's Law Constant | 3227 | 2296 |
| Regression | Kovats Retention Index (standard non-polar) | 79656 | TBD |
| Regression | Log $P$ (experimental) | 22222 | 22222 |
| Regression | Melting Point | 12890 | 5001 |
| Classification | Odor | 2748 | TBD |
| Regression | Odor Threshold (ppm) | 560 | TBD |
| Regression | Optical Rotation | 1,391 | TBD |
| Classification | pH | 955 | TBD |
| Classification | Physical Description | 101,667 | TBD |
| Regression | Refractive Index | 3,699 | TBD |
| Classification | Solubility | 72477 | TBD |
| Regression | Solubility | 72477 | TBD |
| Regression | Surface Tension | 553 | TBD |
| Classification | Taste | 887 | TBD |
| Regression | Vapor Density | 1,343 | TBD |
| Regression | Vapor Pressure (STP) | 6,165 | TBD |
| Regression | Viscosity | 729 | TBD |

# B  Prompt Templates

Figure 3 below shows prompt templates that we designed for property prediction. The table prompt was identical to the format described in and utilized by [20] for the prediction of computed properties.

---

Table Prompt

**Canonical SMILES**
`[START_SMILES]CC(=O)OC1=CC=CC=C1C(=O)O[END_SMILES]`
**Computed Properties**
| Property Name | Property Value
| XLogP3-AA Log P |

---

Zero-shot prompt

Here is a SMILES string for a given molecule:

`[START_SMILES]CC(CC1=CC=CC=C1)NC(C)C2=CC=CC=C2[END_SMILES]`

Question: What is the Molecular Weight?
Answer:

---

Few-shot prompt

Predict the Density of the given chemical compound.
SMILES: `[START_SMILES]CC(C)CCOC(=O)C[END_SMILES]`
Density: 0.87 g/mL
SMILES: `[START_SMILES][H-].[Li+][END_SMILES]`
Density: 0.78 g/mL
SMILES: `[START_SMILES]CC([N+](=O)[O-])(Cl)Cl[END_SMILES]`
Density: 1.43 g/mL
SMILES: `[START_SMILES]C#N[END_SMILES]`
Density: 0.69 g/mL
SMILES: `[START_SMILES]C=CC(=O)OCCC#N[END_SMILES]`
Density: 2.069 g/mL
SMILES: `[START_SMILES]C(=O)(C(=O)[O-])O.[K+][END_SMILES]`
Density:

---

Figure 3: Examples of prompt templates for table, zero-shot, and few-shot prompts.

## C  Statistical testing of experimental vs computed property errors

Here, we describe our procedure for evaluating whether the model has the same error distributions for computed and experimental properties. Overall, we follow the procedure used in the Diebold-Mariano test [41]. However, we made modifications to fit our problem. Diebold-Mariano operates by, given two model forecasts, taking the difference in errors on those forecasts, then standardizing and comparing to the unit normal. In our case, within both experimental and computed property types, there are subgroups of individual properties (e.g., density, Log $P$). These properties often take a different range of values, so errors are not initially comparable. We choose to use absolute value of the difference as our error. To address the scaling problem, we first divide by the standard deviation of the error on individual properties. Then, we take the difference in errors as in Diebold-Mariano. We find the mean standardized difference for computed is -0.413 and for experimental is -0.192, which indicates that ICL more strongly outperforms zero-shot on computed properties than experimental. We speculate this may be due to the inherent property which makes computable properties—computable. Or, it may be because it is far easier to have pretrained on a large quantity of computed properties. We compare the distributions of experimental and computed property standardized error differences using a non-parametric Mann-Whitney U-Test [42, 43]. Using this test,

we find that there is a difference in the location of standardized error distributions between computed and experimental properties with $p = 1.695\mathrm{e}{-5}$.

## D   SMILES length and error

**SMILES length and absolute error are correlated when error is low**   Longer SMILES sequences require more in-depth understanding and must aggregate information across a longer sequence of tokens to make accurate predictions, so these strings should have worse error. See Table 5 for details on each experiment.

Table 5: $R^2$ of SMILES length vs absolute error of predictions, rounded to 3 decimal places.

| Property | Table | Zero-shot | Few-shot (true) | Few-shot (random) |
|---|---|---|---|---|
| Molecular Weight | 0.872 | **0.928** | 0.006 | 0.025 |
| Log $P$ | 0.192 | 0.394 | **0.548** | 0.351 |
| Hydrogen Bond Donor Count | | 0.001 | 0.004 | |
| Hydrogen Bond Acceptor Count | | 0.039 | 0.003 | |
| Topological Polar Surface Area | | 0.106 | 0.016 | |
| Atom Stereocenter Count | | 0.000 | 0.008 | |
| Density | | 0.064 | 0.056 | |
| pK$_a$ | | 0.131 | 0.039 | |
| Melting Point | | 0.153 | 0.008 | |
| Boiling Point | | 0.012 | 0.030 | |
| Heat of Vaporization | | 0.069 | 0.001 | |

# E   Examples of training data in model output

---

**Additional description**

Predict the Heat of Vaporization of the given chemical compound.
SMILES: `[START_SMILES]CCOC=C[END_SMILES]`
Heat of Vaporization: 26.2 kJ/mol
SMILES: `[START_SMILES]C1CCCC1[END_SMILES]`
Heat of Vaporization: 28.52 kJ/mol
SMILES: `[START_SMILES][HH][END_SMILES]`
Heat of Vaporization: 0.90 kJ/mol
SMILES: `[START_SMILES]O=Cl[O][END_SMILES]`
Heat of Vaporization: 30 kJ/mol
SMILES: `[START_SMILES]CN(C)N[END_SMILES]`
Heat of Vaporization: 32.623 kJ/mol
SMILES: `[START_SMILES]CCCCCCCCCCCCCCCC[END_SMILES]`
Heat of Vaporization: 34.2 kJ/mol
Answer:
The heat of vaporization is the energy required to evaporate a liquid at its boiling point.

---

**Additional properties**

Here is a SMILES string for a given molecule:

`[START_SMILES]CCNC1=CC(=NC(=N1)C)N(CCCO)C(C)C[END_SMILES]`

Question: What is the XLogP3-AA Log P?
Answer: -0.5
| Hydrogen Bond Donor Count | 0
| Hydrogen Bond Acceptor Count | 5
| Rotatable Bond Count | 4
| Exact Mass | 210.080336

---

Figure 4: Examples of when Galactica generated extraneous text that reflect training data. The first example comes from ICL with heat of vaporization. In this example, Galactica outputs a memorized sentence describing how heat of vaporization is defined in the real world. The second example comes from zero-shot with Log $P$. In this example, Galactica outputs additional computed properties, such as rotatable bond count, in the pre-trained table format.

