# OpenReview forum: "What a Scientific Language Model Knows and Doesn't Know about Chemistry"
_NeurIPS.cc/2023/Workshop/AI4Science — NeurIPS2023-AI4Science Poster_

### Official Review · Reviewer_ZCEu · 2023-10-17
**title_placeholder**

**Rating:** 6
**Confidence:** 3

**Review:**

Summary

This paper creates a benchmark dataset on molecular property prediction and evaluates Galactica 1.3B on the benchmark. The authors investigates the effects of prompting styles, evaluation data subsets, task categories (computed/experimental molecular properties) on performance. In addition, the authors provide a better understanding of the behavior of in-context learning on the molecular property prediction tasks.


Strength
- This paper performs a range of analysis for molecular property prediction tasks and observed a few interesting phenomenon. Some of which are in contrast to observations made in [1]. This is interesting.
- Open-sourcing benchmark dataset is useful!


Weakness
- My main concern is the paper mentions "understanding" quite a few times. The definition of "understanding" is mentioned in second paragraph of introduction and I don't find it particularly satisfying. It seems "understanding" mentioned in the paper is a catch-all phrase that describes the ability of language model to generalize to unseen data points. The authors should just say we are interested in the generalization of language model. If there are subtle points that is more nuanced than generalization, the authors might want to clarify that.



Additional Comments
- [2] is a relevant reference that should be included in related work and discussed. This paper would also help with authors' understanding of the meaning of "understanding in-context".
- Table 1. Label what rows/columns are for clarity, e.g., "Prompt Style" for columns and "Task (Data Subset)" for rows.
- It would be useful to get error bars for valueds in Table 1, e.g., random sample data subsets many times then compute the scores based on different prompting strategies over these subsets. It would be useful to have a less noisy estimate of model property in Table 1, e.g., use paraphrases of the prompt for evaluation. As mentiond in the text too, minor variation in prompt format changes evaluation performance significantly. It's challenging to draw conclusions when the values couldn't be fully trusted.
- In Section 4.1, what is the reason for different eval performance when evaluated on "first 100" vs. "random" compounds? Explain a bit.
- In Section 4.1, would the authors explain a bit more why zero-shot better than few-shot(true) better than few-shot(random) imply that LLM use prior memorized knwoledge to make predictions? The language model would resort to its memory to retrieve the correct values for molecular weight and logP regardless of the prompting styles used. It's not true that providing in-context examples prevented the language model from recall from memory.
- In Section 4.4, what does "understanding of the correct output" mean?
- In section 4.4, model generating unrealistic values is not related to memorization. Remove the sub-section or put it in a better location.
- Galactica imploys prompt pre-training and is not instruction tuned. Some issues mentioned in the paper, e.g., high variance in performance due to prompting styles, worse zero-shot performance, generation of memorized additional properties, can all be potentially alleviate with instruction tuning (perhaps using domain-specific instruction tuning dataset).
- You could argue memorization is the best a LM can do, since the ground-truth molecular property is determined via experiments or via step-by-step calculations that is not taught to the LM. It might be useful to do finetuning on instruction datasets that includes step-by-step calculation of how to compute the computed properties.

References
1. Rethinking the Role of Demonstrations: What Makes In-Context Learning Work?
2. What In-Context Learning “Learns” In-Context: Disentangling Task Recognition and Task Learning

---

### Official Review · Reviewer_GR5b · 2023-10-22

**Rating:** 7
**Confidence:** 4

**Review:**

**Summary**

The recent release of LLMs have led to many works exploring their potential for chemistry applications. This work performs a systematic investigation into the extent to which Galactica (a scientific LLM) "learns" chemistry when trained on a large corpus. Through various prompting methods and predictive tasks, the authors provide insights into limitations of directly applying LLMs, such as the predictive sensitivity to prompting. The insights in this work are especially pertinent for practical applications.

**Main Review**

The paper is well written and easy to follow. Given its submission to the Attention Track and the popularity of LLMs, insights into the behavior of LLMs for chemistry is interesting and useful to practitioners and researchers.

**Strengths**

* Interrogating the predictive performance under various prompting templates is done in a systematic way and sheds some insight into sensitivity of the output predictions. The errors for molecular weight and logP are quite large and exhibit high variance but it is interesting that Galactica seems to have some notion of "similarity" (as measured by the Levenshtein distance) to known examples. This is important if one wishes to use in-context learning or fine-tuning for chemistry applications as acquiring data can be expensive (such as a wet-lab experiment) and data can be noisy.

* The ICL experiments interrogating "% copied" and the tendency to output the tokens towards the end of the context have important implications for practical applications.

**Limitations**

* While the choice to use Galactica was motivated, it would be more conclusive if other LLMs were used, such as GPT >= 3.5.

* LLMs can give different output given the same prompt. For certain properties, small discrepancies can have large implications, for example in the pKa example cited by the authors. Was replicate querying studied in this work?

**Questions**

1. In the experiments that interrogate accuracy based on position of compounds in the dataset, it was stated that randomly selected compounds have lower error than the first 100 compounds. Is the assumption that only the earlier compounds were used for training? Or they would be more likely to be "memorized"?

2. Table 1 labels can be more clear: The “random” for property means randomly sampled from PubChem database while the “random” in Few-shot means the property value.

3. In the Appendix, more properties were extracted than shown in the results. It would be nice to include the entire dataset to draw more conclusive findings. Or were these properties not investigated due to smaller dataset sizes?

Overall, while performing the analysis on other LLMs would make the work more holistic, the question being studied is especially relevant for the Attention Track. The insights are useful to practitioners and for future LLMs for chemistry research.

---

### Meta-Review · Area_Chair_qAht · 2023-10-26

**Recommendation:** Accept (Poster)
**Confidence:** 3

**Metareview:**

The paper explores a timely question of understanding how much chemistry do LLMs understand. An important contribution of the paper is a benchmark consisting in molecular property prediction tasks presented in a zero or few shot manner. A shortcoming is the focus on Galactica 1.3B, without showing results for frontier LLMS (GPT-4/Claude-2/PALM), as pointed out by the reviewers. Another area for improvement is that the questions focus on the task of predicting molecular properties, rather than more generic questions probing chemical understanding. It is also somewhat challenging to contextualize results without some baselines (such as simple machine learning models). All in all, it is my pleasure to recommend acceptance of the paper.